# The Effect of the Synthetic Procedure of Acrylonitrile–Acrylic Acid Copolymers on Rheological Properties of Solutions and Features of Fiber Spinning

**DOI:** 10.3390/ma13163454

**Published:** 2020-08-05

**Authors:** Ivan Y. Skvortsov, Elena V. Chernikova, Valery G. Kulichikhin, Lydia A. Varfolomeeva, Mikhail S. Kuzin, Roman V. Toms, Nikolay I. Prokopov

**Affiliations:** 1A.V. Topchiev Institute of Petrochemical Synthesis, Russian Academy of Sciences, Leninsky Av., 29, 119991 Moscow, Russia; chernikova_elena@mail.ru (E.V.C.); klch@ips.ac.ru (V.G.K.); varfolomeeva.lidia@mail.ru (L.A.V.); gevahka15@gmail.com (M.S.K.); 2Department of Chemistry, M.V. Lomonosov Moscow State University, Leninskiye Gory, 119991 Moscow, Russia; 3M.V. Lomonosov Institute of Fine Chemical Technologies, Russian Technological University, Vernadsky Av., 86, 119991 Moscow, Russia; toms.roman@gmail.com (R.V.T.); prokopov@mitht.ru (N.I.P.)

**Keywords:** acrylonitrile, acrylic acid, radical polymerization, reversible addition–fragmentation chain transfer, solutions, rheology, fiber spinning, fibers

## Abstract

The influence of introducing acrylic acid (AA) into the reaction mixture with acrylonitrile at the synthesis of copolymers by free-radical polymerization (FRP) and radical polymerization with reversible addition–fragmentation chain transfer (RAFT) on the rheological properties of their solutions in dimethyl sulfoxide, as well as on the capability to spin fibers by the mechanotropic method, is analyzed. The influence of AA dosing conditions on the rheological properties of the solutions in the concentration range above the crossover point was not revealed. In the case of RAFT synthesis, the rheological properties differ distinctively in the high concentration region that is expressed by unusual viscoelastic characteristics. Dilute solution viscometry revealed the influence of the comonomer loading order on the interaction intensity of the copolymer macromolecules with a solvent, which is more pronounced for samples synthesized by FRP and can be associated with the copolymers’ molecular structure. Fiber spinning from solutions of polyacrylonitrile and its copolymers (PAN) synthesized by the RAFT method was not able to achieve a high degree of orientation drawing, while for polymers with a wider molecular weight distribution synthesized by FRP, it was possible to realize large stretches, which led to high-quality fibers with strength values up to 640 MPa and elongation at a break of 20%.

## 1. Introduction

Polyacrylonitrile and its copolymers (PAN) are the most popular polymers for obtaining not only wool-like textile fibers, but also high-strength and high-modulus carbon fibers [1]. The process of producing carbon fiber (CF) consists of several stages, each of which has a significant effect on the properties of the final CF: the chemical composition and characteristics of the copolymer, preparation of the dope, spinning of the precursor fiber and its subsequent oxidation and carbonization [2]. To date, a sufficient number of studies in the available literature have been devoted mainly to only one of the process stages. For example, it is polymer synthesis [3,4,5,6,7], precursor fiber spinning [8,9], thermal stabilization of the PAN precursor [9,10,11] and, finally, PAN carbonization [12,13].

Significantly fewer publications have been devoted to comprehensive studies of the process of carbon fiber formation: from the stage of copolymer synthesis to the production of technical threads and CF from them. A rare exception are the following papers: [14] which presents data on the synthesis of PAN by the method of reversible addition–fragmentation chain transfer (RAFT), preparation of precursor fibers from its solutions and, further, CF; and [15], devoted to the synthesis of PAN in supercritical CO_2_, with its dissolution in dimethylsulfoxide (DMSO) and the precursor fiber spinning process carried out by the wet method followed by carbonization. The authors of these works emphasize the importance of a comprehensive review of the various stages of the process, stating that each of them has a significant impact on the subsequent, and ultimately on the structure and properties of the final product.

PAN homopolymer is rarely used to produce carbon fibers due to the strong interactions of the nitrile groups of macromolecules [16,17]. Instead, double and ternary copolymers are used with a mass content of comonomers, typically not higher than 10% [18,19,20]. In the process of synthesis in the solution, the main attention is paid to the polymer yield, temperature, process time, content of by-products, and filtration. At the same time, the analysis of the molecular characteristics of the polymer is often limited to the molecular weight and molecular weight distribution (MWD) and, rarely, to the average composition of the copolymer, while the character of the comonomer distribution along the chain is estimated only by the constants of their activity during synthesis. The use of direct methods, for example, NMR for PAN copolymers, is limited due to the low content of comonomers and, therefore, the low probability of their uniform entry into triads, pentads, etc. Nevertheless, it was shown in [16,21] that compositional heterogeneity, along with the molecular weight and MWD of the copolymers specified at the synthesis stage, affects the morphology and structure of the spun fibers and their behavior during thermal stabilization.

In general, during thermo-oxidative stabilization, cyclization of nitrile groups occurs, which is initiated by temperature, atmospheric oxygen, a residual initiator, and proceeds according to the radical mechanism. In the case of homopolymers, this stage is accompanied by an intense exothermic effect [22], which can cause destruction of macromolecules. However, in the presence of monomers containing acid groups, a partial realization of the ionic mechanism is possible; in this case, cyclization begins at a lower temperature, proceeds in a wider temperature range, and the total exo-effect decreases [23]. However, in addition to the presence of comonomers, the compositional heterogeneity of the polymer can also influence the structure and rheological properties of solutions that determine their behavior during spinning.

The structure and morphology of the resulting fibers, which are the most important factors in subsequent transformations during thermolysis, depend to a large extent on the chosen spinning method. Only in rare cases is it possible to obtain fibers uniform in cross-section with wet spinning [15,24,25]. First of all, this is due to mass transfer processes occurring in the coagulation bath. In the case of using a sufficiently rigid coagulant, which causes a large difference in osmotic pressures, a rapid gelation of the surface of the solution jet occurs, leading to the formation of a dense shell and a fairly loose fiber core (shell–core effect). In addition, during drawing, the shell can break and the coagulant can penetrate into the resulting cavities, which leads to heterogeneous morphology, surface defects, and, accordingly, to reduced mechanical properties [26].

Much less is known about the role of the viscoelastic properties of dopes in spinning stability and fiber quality. As an exception, we refer to the work [27], which demonstrated the decisive influence of rheological properties when spinning thermal-resistant fibers from gel-like solutions of the heterocyclic polymer—polyaminonaphthoylenimide.

An alternative to the methods of fiber spinning from polymer solutions using coagulants is the mechanotropic spinning method that we are developing [9], which, on the one hand, resembles electrospinning [28] and, on the other hand, classical dry spinning. During mechanotropic spinning, the mechanism of phase separation in the solution jet caused by large longitudinal deformations is realized. In contrast to electrospinning, the tensile force is not the electric potential difference between the die and the collector, but an intense mechanical field.

At the same time, in contrast to the evaporation of the solvent from the forming jets at a high temperature in the dry spinning method, during the mechanotropic process, diffusion of the separated solvent phase onto the surface of the forming jet/fiber proceeds with the subsequent formation of droplets and their removal either mechanically or as a result of evaporation. In this spinning method, the role of the viscoelastic properties of the solution becomes more defined, because phase separation is based on the concept of elasticity of the spinning solution. Thus, while in the case of wet spinning, the decisive factor is the mass transfer interaction of the solution with the coagulant with the “subdued” effect of viscoelasticity, in mechanotropic spinning, the rheological properties of the spinning solution come to the fore.

In this paper, the rheological aspects of the behavior of solutions, as a necessary basis for the mechanotropic spinning process, will be discussed in detail with examples of solutions of acrylonitrile copolymers synthesized using various methods that lead to the production of polymers with different distribution of units in the chain, heterogeneity in molecular weights and composition. Previously, a detailed combined study on the influence of these factors on the rheological properties of solutions, the stability of mechanotropic spinning, and the properties of the resulting fibers was not carried out.

## 2. Materials and Methods

### 2.1. Materials

Acrylonitrile (AN) (99%) and acrylic acid (> 99%) of the “Acros Organics” (Fair Lawn, NJ, USA) were used in the synthesis of double copolymers. Dinitrile azo-bis-isobutyric acid (AIBN) and anhydrous potassium persulfate (PSK > 98%) purchased from “Aldrich” (St. Louis, MO, USA) were used as an initiator. As the RAFT agent, 2-(dodecylthiocarbonylthioylthio)-2-methylpropionic acid(C_12_H_25_–SC(=S)S–C(CH_3_)_2_COOH, CTA) from “Aldrich” (St. Louis, MO, USA) was used.

The polymerization was carried out in an argon atmosphere in a 100 mL three-necked flask equipped with an overhead stirring device with an anchor type mixer. A solution containing the calculated amount of initiator and RAFT agent was placed in the flask and AN was loaded. Acrylic acid (AA) was added in the following order: (1) continuously at a rate of 0.32 mL/h; (2) semi-batch, i.e., after a certain time from the beginning of the synthesis, at a speed of 0.63 mL/h; and (3) the whole amount at the beginning of the synthesis. At a given time, samples were taken to determine the conversion, molecular weight characteristics and composition of the copolymers. In all syntheses, the mass ratio of AN and AA was 90:10. RAFT copolymerization was carried out at 55 °C and the total concentration of monomers in the solution was 40 wt.% (Series A) [6,7]. Free-radical polymerization (FRP) was carried out at 65 °C and the total mass content of monomers in the solution was 22% (Series B).

The molecular weight of AN copolymers was measured by gel permeation chromatography (GPC) on an Agilent Technologies PolymerLabs GPC-120 chromatograph (Santa Clara, CA, USA) in dimethylformamide containing 0.1 wt.% LiBr at 50 °C. MW was calculated according to PMMA standards and converted to polyacrylonitrile according to known coefficients. The characteristics of the obtained copolymers are presented in Table 1. The detailed information regarding the molecular weight characteristics of the copolymers and their composition throughout the polymerization process are given in the Appendix A, which fully explain the nature of the differences among the copolymers.

Glass vials with hermetically sealed caps were used to prepare and store of solutions. Depending on the polymer concentration, various methods of preparing solutions were used. Solutions with concentrations up to 5% were prepared using a magnetic stirrer and used to measure the intrinsic viscosity and rheological properties by rotational rheometry. Dissolution proceeded during a 24 h period at 50 °C. High viscous solutions with concentrations above 5% were prepared using a paddle mixer with a J-shaped rotor. Mixing was carried out with a rotor speed of 60 rpm for 24 h at 70 °C. Dopes for spinning with concentrations of 22 and 25% were prepared at a rotor speed of 10 rpm for 72 h at 70 °C.

### 2.2. Methods

#### 2.2.1. Rheology

The rheological behavior of the solutions was investigated in continuous and oscillatory modes of shear deformation on an Anton Paar MCR 301 (Austria) rotational rheometer at temperatures from 20 to 80 °C using a cone-plate operating unit with an angle between the cone and the plate of 1°.

For solutions with concentrations less than 10%, a cone with a diameter of 50 mm was chosen; for concentrations of more than 10%, a cone with a diameter of 25 mm was applied. The following characteristics were obtained: flow curves in the range of shear rates of 10^−2^–10^4^ s^−1^ and frequency dependences of the storage and loss moduli in the linear viscoelastic region at frequencies from 0.6 to 628 rad/s.

The intrinsic viscosities of the solutions were measured on a Ubbelode viscometer at a temperature of 25 °C in accordance with ASTM D2857 [29].

#### 2.2.2. Modeling of Mechanotropic Spinning

To assess the features of the behavior of jets under uniaxial tension of solutions, an apparatus was used, which made it possible to stretch a drop of dope at a given speed to a constant length [9,30]. Solutions with concentrations of 10 and 15% were investigated using a bright-field lighting option, which allows achieving maximum contrast in the display to visualize fine fibers and droplets of the solvent released. For a concentrated 20% solution, an original illumination system was designed to visualize the phase separation process during jet stretching.

The experiments were carried out according to the following scheme:A solution drops with a volume of ~10 μL was squeezed from a syringe located at the bottom of the unit;By displacing the movable syringe, a drop was brought into contact with the upper plate of the optical fiber lens;The droplet was stretched at a constant speed of 65 mm/min by linearly moving the syringe to a distance of 11 mm. The duration of the process was approximately 10 s.

At the initial moment of droplet stretching, the video recording process was started at 60 frames per second on a ToupTek XFCAM1080PHD camera (ToupTek Photonics Co, Hangzhou, China), paired with LOMO 4 × or 10 × (for 10 and 15% solutions) lenses (for 20% solutions), which renders it possible to obtain images with a resolution of up to 5 and 2 μm, respectively.

#### 2.2.3. Fiber Spinning

Previously [9], a method of continuous mechanotropic spinning from a die/capillary with one hole was developed. In this work, the fiber spinning was carried out on a laboratory line from a die with 10 holes with a diameter of 500 μm. The spinning fibers underwent two stages of orientational drawing in air, followed by washing with water from the residual solvent, drying and thermal drawing at 100 °C. The spinning line scheme is shown in Figure 1.

#### 2.2.4. Fiber Characterization

The diameter of each fiber was averaged from ten measurements at different location along the fiber length. To perform these measurements, the Biomed 6PO optical microscope equipped with a ToupTek XFCAM1080PHD camera (magnification of 60× and an accuracy of ±0.3 μm) was used. The inhomogeneity of fiber thickness was characterized by the difference between the maximum and minimum values of the fiber diameter.

Mechanical properties of the fibers were measured on an Instron 1122 Tensile machine with a basic filament length of 10 mm and an extension speed of 10 mm/min. All measurements were performed at 23 ± 2 °C. The reported results were averaged for at least 10 tests.

The experimental data that exceeded 3σ were excluded from the analysis. Confidence interval was estimated with a 95% confidence level. Standard deviation was calculated as $\bar{x}-1.96\frac{\sigma }{\sqrt{n}}$. The sample average was calculated as: $\bar{x}=\frac{\sum_{i=1}^{n}x_{i}}{n}$, where *n* is a sample of observations and *σ* is a sample standard deviation, defined as the square root of the sample variance: $\sigma =\sqrt{\frac{\sum_{i=1}^{n}{\left(x_{i}-\bar{x}\right)}^{2}}{n-1}}$.

## 3. Results and Discussion

### 3.1. Rheology

From general considerations, it follows that the most clearly compositional heterogeneity of PAN macromolecules should be manifested in the study of dilute solutions. This thesis for similar polymers was previously confirmed in [31]. In continuation of this approach, the properties of the dilute solutions were analyzed by capillary viscometry. Figure 2 shows the dependences of reduced viscosity on concentration for all the studied polymers.

For series B, these dependences have close increment values, which indicates the absence of a pronounced effect of the distribution of comonomer units on the flow mechanism of the dilute solutions when there is no interaction between polymer coils. For all the samples of series B, the intrinsic viscosity is almost the same.

In frames of series A samples, the reduced viscosity is maximal for sample A3 and minimal for A2. In the same sequence, a change in the slope characterizing the interaction of the polymer with the solvent is observed. The concentration dependence of the reduced viscosity for sample A2 is below the corresponding dependences for all samples of series B, and for sample A1 it is in the same region.

These dependences were analyzed using the Huggins and Martin equations:(1)$$\eta_{\mathrm{sp}}/c=\left[\eta \right]+{\;K}_{H}{\left[\eta \right]}^{2}c$$
and
(2)$$\ln(\eta_{\mathrm{sp}}/c\left[\eta \right])=K_{M}\left[\eta \right]c$$
where η_sp_/c is the reduced viscosity, [η] is the intrinsic viscosity, and K_H_ and K_M_ are the Huggins and Martin constants, respectively. The results of calculations of the values of the intrinsic viscosities of the polymers, as well as the values of the Huggins and Martin constants, are shown in Table 2.

It can be observed that the molecular weight values for samples A1 and A3 do not correlate with the viscometric data of dilute solutions. This indicates the different sizes of the molecular coils, i.e., under the conditions of continuous comonomer introduction, the coil is more compact than in the case of a batch loading. Based on these data, it can be assumed that the chain of copolymer A1 is more flexible than the chain of copolymer A3. Sample A2, obtained by the method of semi-batch introduction of AA in the reaction mixture, has the lowest molecular weight and intrinsic viscosity and it is characterized by the worst polymer affinity for the solvent (when evaluated through the Huggins constant) among all the copolymers studied.

Based on general considerations, it can be assumed that the RAFT copolymers have approximately the following AA distribution in the chain: A1 (continuous addition) is the most uniform; A2 (semi-batch loading) is the block-random, comprised of homo-PAN and random copolymer; and A3 (batch loading) is the most irregular. The obtained data on the viscometry of dilute solutions, in principle, do not refute the stated considerations.

Comparing the data regarding the dilute solutions of series A and B, we can conclude that the method of introducing the additive does not have a clear effect on the nature of the interaction of the copolymer with DMSO in the case of FRP, while in the case of the RAFT process, there are slight differences. The latter can be due to both differences: in chain flexibility and in the molecular weights of the synthesized samples.

Presentation of the study results of the dilute solutions in the coordinates of the Martin equation (Figure 3a) allows us to estimate the intensity of the polymer—solvent interaction when using the volume engaged by macromolecules as an argument, i.e., c[η]. In this case, there are no significant differences between the series and the reduction in data, as the volume of macromolecules is already almost sufficient. Taking into account the values of the Martin constant reduces the hydrodynamic properties of the dilute solutions of all polymers to almost the same general dependence (Figure 3b).

Turning to the analysis of the rheological behavior of the concentrated solutions, we consider the flow curves of solutions of different concentrations obtained for polymers synthesized by FRP (Figure 4).

Regardless of the loading method of the comonomer, the rheological behavior of the solutions is similar in the entire range of concentrations studied. This means that the shape of the flow curves remains traditional (except for a 20% solution): the branch of the Newtonian viscosity and the structural branch. Small differences in the absolute values of viscosity in the region of high concentrations may be due not to compositional heterogeneity, but to a difference in the molecular weight of the samples. For a 20% solution, it is not possible to reach the Newtonian flow region at low shear rates, which can be explained by its increased structuring.

Figure 5 shows the viscoelastic properties of moderately concentrated copolymer solutions (10% and 20%) synthesized by FRP.

If, for 10% solutions, the differences in the behavior of individual samples of the series are minimal, then at a concentration of 20%, a tendency to decrease in both moduli can be detected as irregularity of the loading of acrylic acid increases (from sample B1 to sample B3).

Flow curves for a series of copolymer solutions obtained by the RAFT method are presented in Figure 6.

Significant differences in rheological behavior of the solutions of different samples of the same concentration were observed. Two factors should be noted that can affect the viscosity properties. Firstly, these are differences in molecular weights. Secondly, this is the order of loading the comonomer into the reaction medium, which a priori should lead to a more uniform distribution in the case of A1 and the block-statistical in the case of A2. The solutions of copolymer A3 have a maximum viscosity at all concentrations, and that of sample A2, a minimum. Samples with a higher molecular weight (A1 and A3) at a concentration of 20% in the studied range of shear rates do not exhibit constancy of viscosity at low rates. The reason for these differences is probably due to the combined effect of both factors: the difference in the molecular weights of the samples and the difference in the distribution of comonomers along the chain.

Based on the obtained data, the dependences of the Newtonian viscosity on concentration for all the studied polymers were plotted (Figure 7). They are of a usual form, generally described by two power-law equations with exponents ~0.5 (before c*) and ~5–6 (after c*). The critical concentration (c*) reflects the transition from the dilute solutions with non-interacting macromolecules to semi-diluted, in which contacts between macromolecules arise as well as their interpenetration.

The critical concentration separating the regions of dilute and concentrated solutions practically does not depend on the order of acrylic acid addition into the reaction system or on differences in molecular weight, and it is equal to ~2%. The viscosity of solutions of the B-series copolymers both in the field of dilute and concentrated solutions practically does not depend on the order in which the comonomer is introduced. A different picture appears for the solutions of series A copolymers. The viscosities of samples of the A3 solutions exceed the corresponding values of the A1 samples by half an order of magnitude over the entire concentration range, which may be due to the difference in the distribution of comonomer units in the chain. The solutions of A2 copolymers have a lower viscosity, apparently due to lower molecular weight and specific (microblock) compositional heterogeneity.

When considering the viscoelastic properties of the concentrated solutions (Figure 8) of series A samples, it can be seen that for both concentrations of the copolymer solution, the maximum values of the moduli occur for sample A3, and the minimum for sample A2. In addition, for a 20% solution of sample A3 in the terminal zone, the elastic and loss moduli practically coincide, which indicates that the tangent of mechanical losses is equal to unity, thus explaining the high elasticity of the solution. It is also important that significant differences appear in the region of characteristic relaxation times of polymer chains corresponding to the intersection of the dependences G’ (ω) and G” (ω) [32], which suggests variation in macromolecular rigidity. In general, differences in rheological behavior may be due to specificity of acid group distribution in copolymers and different chains length.

In terms of mechanotropic spinning, an important characteristic is the frequency at which the elastic reaction begins to exceed the dissipative one. Using the Cox–Merz rule [33], one can estimate the shear rate corresponding to the crossover (the intersection of the frequency dependences of the elastic and loss moduli), above which elasticity becomes the predominant response of the solution. These data are shown in Table 3.

As is seen from the table, the minimum frequency corresponding to the crossover for both concentrations of copolymer solutions of series A is observed for the solution of sample A3, and the maximum for the solution of sample A2. At the same time, these frequencies do not differ significantly for the solutions of series B copolymers. It makes sense to discuss these data together with the analysis of the exponents in power-law dependences of moduli on frequency (Figure 9).

According to classical concepts [34], in the low-frequency region, i.e., in the linear region of viscoelasticity, this exponent should be equal to 2 for G’ and 1 for G”. However, in our case, this rule is not fulfilled, especially for G’, and the difference between theory and experiment increases with increasing concentration of the solution. Thus, for 10% solutions (Figure 9a), the maximum slope of the log G’ (log ω) dependence reaches ~1.7 for the solutions of polymers A2 and B1 and only ~1.4 for the same sample A2 in 20% solutions (Figure 9b). In addition, the lowest and equal in magnitude values of the elastic and loss moduli for 20% solutions of sample A3 are registered. In this case, the solutions look like a gel, which correlates with the minimum crossover frequencies. The reasons for this behavior are not yet clear, but below we will return to a discussion of this anomaly.

The data on the Newtonian viscosity dependences of 10% solutions at low shear stresses on temperature are presented in Figure 10. An analysis of the temperature dependences of viscosity did not reveal significant changes in the rheological behavior, since practically for the solutions of all samples the flow activation energy calculated by the Arrhenius equation is close to 26 kJ/mol.

The same data demonstrate the significant role of the acrylic acid loading order into the reaction mixture for polymers synthesized by the RAFT method. Thus, for samples A1 and A3, the molecular weight values are close, but the viscosity of the copolymer A3 solution is more than an order higher than the viscosity of the corresponding A1 solution. The solution of copolymer A2 has the lowest viscosity in this series, for which the molecular weight is lower than in previous cases and, in addition, it implements a specific, block-statistical distribution of comonomer units. Moreover, for the solutions of sample A3, the maximum viscosity values were also recorded (Figure 6), as well as the elasticity and loss moduli (Figure 8), which fits into the overall picture of the increased compositional heterogeneity of the copolymer with the simultaneous introduction of the second component.

At the same time, it should be noted that for the solutions of series B samples, the loading order does not significantly affect either the viscosity values or the characteristics of viscoelasticity. From this, a preliminary consideration can be made that the RAFT process is more sensitive to the order of introduction of the second component in the synthesis of the copolymer compared to FRP. Apparently, in the case of series B, the set of chains with different microstructures levels each other, and in the case of the RAFT, the different distribution of links along the chain is determined by the order of introduction of AA into the synthesis.

### 3.2. Modeling of the Mechanotropic Spinning

Preliminary experiments to simulate mechanotropic spinning were carried out by stretching a drop of a solution to a constant length to assess the solutions spinnability. Its ability to form fibers varies significantly depending on copolymer content in the solution. In the concentration range close to the crossover point of the concentration dependence of the viscosity, corresponding to the appearance of entanglements (~5%), the formation of a thin filament with a diameter of ~1 μm—which breaks after 0.1–0.5 s of forming thin fibers about 1 cm long—takes place. In this concentration range, most of the solvent flows into domains of low pressure (thickening parts at the beginning and the end of the stretched jet). The phenomenon of the onset of fiber formation in this field of concentration is described in more detail for various solutions of AN copolymers in [35].

With an increase in copolymer content in the solution up to 10% (Figure 11a), the drop to be stretched for 1–5 s and 5–10 cm long fibers is obtained. Here, the initial rapid thinning of the liquid filament, the release of solvent by the entire length of the jet/fiber in the form of beads on its surface and coalescing over time into larger droplets are observed. In this case, the thickness of the fiber does not depend on the rate of extension of the droplet, remaining within ~2 μm.

With an increase in the solution concentration to 15% (Figure 11b), stabilization of the lifetime of an elongated drop to 10–60 s occurs. The resulting fiber is not stable and breaks easily under the influence of natural weak air breath. The thickness of the obtained fiber from a 15% solution remains in the range of 3–5 µm and does not depend on the stretching rate.

Upon transition to a 20% viscoelastic solution (Figure 11c), the process undergoes qualitative changes. Due to the high viscosity, the initial thinning of the liquid filament occurs within tens of seconds and ends when diameters of the order of 200 μm are reached when stretched to a constant length, followed by turbidity in the central part of the stretched jet, which propagates to the entire length of the liquid filament. Turbidity is caused by phase separation, which ends with the release of solvent to the surface, moving either in the region of lower Laplace pressure (up) or under the action of gravity forces (down).

Additional stretching of a drop of such solution in the domain of irreversible strains until the start of phase separation allowed us to vary the diameter of the resulting gel fiber over a wide range: from 5 to 100 µm, depending on the stretching rate. Thus, for this group of copolymers, the most optimal from the viewpoint of implementing mechanotropic spinning is an extremely high concentration of the viscoelastic solution.

The tensile strain rate can be calculated by the following equation [9]:(3)$$\dot{\varepsilon }\left(z\right)=-\frac{2Q}{\pi a^{3}}\frac{da}{dz}$$
where *Q* is the flow volume rate, a is the radius of the jet, and *z* is the distance from the die. Substituting the values of the typical flow rate and jet thickness in the thinning zone to a constant value characterizing the transition from a liquid solution to a gel-like fiber, we obtain $\dot{\varepsilon }$ of the order of 1–10 s^−1^, which corresponds to the crossover point values on the frequency dependences of the moduli for these solutions (Figure 12). By this way, it is possible to select the necessary and sufficient viscoelastic parameters for stable mechanotropic spinning.

### 3.3. Fiber Spinning

For the preparation of dopes, the polymer concentration was selected based on their viscoelastic properties: the solution is able to flow at small deformations (at the exit of the die) and it transfers into the elasto-viscous region (above the crossover frequency), thereby preventing the occurrence of Rayleigh instability.

This made it possible to obtain fibers of a given thickness by selecting a specific spin-bond hood. It corresponds to solutions located near the crossover point of moduli. Based on this, a 22% concentration was selected for copolymer A1 and 25% for A2 and all the solutions of series B. Flow curves for these solutions of both series at 70 °C are shown in Figure 12a. For the solutions of the A3 copolymer with the most irregular distribution of comonomer units in the chain, dependences of dynamic moduli for different concentration of solutions are shown in Figure 12b.

At room temperature, these systems in almost the entire considered shear rate and frequency ranges are viscoelastic media with a high viscosity, which decreases by three orders of magnitude with an increase in shear rates. Heating to the spinning temperature (70 °C) leads to a decrease in the elasticity of the solutions (see Figure 12c). Based on the data in Figure 8 and Figure 12b, 20% solutions of copolymer A3 are characterized by a significant degree of structuring, which follows from the low values of the exponent in the elastic modulus on frequency dependence equal to ~1, and almost complete coincidence of the sections of the dependences G’(ω) and G”(ω) in the low frequency region. It also appears rather strange that for the dependence G”(ω), the slope in the terminal zone is equal to the traditional value close to 1. From the point of view of dissipative properties, a 20% solution of copolymer A3 is in the region of linear viscoelasticity, while their elastic properties correspond to a strongly nonlinear region.

The fibers were spun by the mechanotropic method [35] from the solutions described above (except the solutions of sample A3). The phase separation in the solution is achieved due to large tensile deformations and proceeds without the use of a coagulant. In the case of the copolymer A3 solutions, the spinning was unstable, therefore, high-quality fibers could not be obtained.

Since the structure and properties of the fibers (when choosing the optimal rheological properties of spinning solutions) are determined by the various stages of drawing [36], it makes sense to consider them in more detail. At the stage of spin-bond hood, the phase separation begins and a jet of a viscoelastic solution transforms into a just-spun fiber [10]. Up to this point, in the zone of the flow of the solution from the die, it is easily deformed, which makes it possible to adjust the thickness of the final fiber.

The maximum draw ratio of plasticizing stretching of the just-spun fiber in air (V2/V1) for the continuous spinning process carried out on solutions A1 and A2 was ~1.2, while the jets of the solutions obtained from copolymers of FRP B1–B3 were stably stretched at a more-than-doubled ratio. For the orientation drawing, the maximum ratios were chosen to achieve the highest degree of molecular orientation. The limiting thermal drawing ratios at 100 °C for both systems were 2.3.

Fiber spinning conditions are presented in Table 4.

As a result, the total draw ratio during the spinning of fibers from the copolymers of the A1 and B1 solutions was two times different in favor of a solution of polymer B1, and it was shown that different deformability of the jut-spun fibers was initiated at the stage of plasticizing drawing.

Microphotographs of the fibers shown in Figure 13 for all samples indicate their sufficiently smooth surface without visible defects.

The decisive characteristics of the further use of the obtained fibers are mechanical properties, as presented in Table 5.

The total draw ratio realized during spinning has a direct effect on the strength of the fibers. For this reason, the strength of fibers from series B copolymers is higher than for series A copolymers. However, the elastic modulus of fibers is almost the same for copolymer samples of different synthetic methods, despite a difference of twice the amount in the total draw ratio. The difference between copolymers consists not only in compositional heterogeneity, but also in other factors: molecular weights, polydispersity and, possibly, branching of macromolecules.

Macromolecules with a high molecular weight are more difficult to orient during spinning, and it is challenging to give them the necessary mobility for reorganization. For such polymers, the spinning solutions should be less concentrated than that used for sample A1, which had a lower strength compared to the other copolymers. An interesting behavior is demonstrated by copolymer A2, which has a specific, block-statistical chain structure. Its solutions are well formed, and the strength properties are at the highest level. In sample A3, microheterogeneity is so high that the dope is close to the gelation stage and, possibly, for this reason, the elasticity and loss moduli at low frequencies are almost the same. In addition, with increasing frequency, G’ becomes higher than G” (Figure 5 and Figure 12).

Furthermore, the intrinsic viscosity of the solutions of series A samples is extremely sensitive to the order of loading of the comonomer, which may be associated not only with the nature of the distribution of acrylic acid units in the chain, but also with differences in their molecular weights and, possibly, the degree of branching. Another reason for the increased structuring of the solutions of the A3 sample may be the increased concentration of acid groups in terminal parts of the chain, which can cause the formation of H-bonds and an “artificial” increase in molecular weight as a result of non-covalent binding of macromolecules. The planned study of these samples by NMR spectroscopy will clarify the reality of this event. For series B samples, this effect was not observed.

The obtained precursor fibers are prepared for thermal stabilization and carbonization processes.

## 4. Conclusions

The influence of the loading order of acrylic acid into the reaction zone on the rheological properties and spinnability of double co-PAN solutions for FRP and RAFT processes was studied. Assuming a priori that the loading order predetermines the compositional heterogeneity of the macromolecules between themselves (in the case of FRP) and along the chain (RAFT process), we demonstrated its manifestation for the second version of the synthetic procedure. Thus, the rheological properties of the concentrated solutions in the case of RAFT polymerization “feel” compositionally heterogeneous, while those of FRP differ only in dilute solutions to a small extent. The solutions of samples obtained by the RAFT method with equal molecular weight have a lower viscosity when compared with equi-concentrated solutions of samples prepared by FRP, regardless of the nature of the distribution of the comonomer along the chain.

From polymer solutions with the supposedly the most uniform distribution of comonomer units along the chain, fibers were spun by the mechanotropic method. It was found that the spun fibers from the solutions of AN–AA copolymers obtained by RAFT synthesis are less capable of orientational drawing in comparison with the corresponding copolymers obtained by the standard radical polymerization. For this reason, for PAN copolymers obtained by FRP, it is possible to achieve a higher degree of molecular orientation during spinning. This mainly affects the fiber strength, which reaches 700 MPa for the copolymer obtained by FRP, and only 500 MPa for the copolymer synthesized in the RAFT process. In this case, both families of fibers have the same values of elongation at break of about 20–30% and an elastic modulus of 7 GPa.

A possible reason for such a difference in the orientation abilities of the spinning fibers can be differences in both the distribution of comonomers along the chain and in the values of molecular weights and MWD.

## Figures and Tables

**Figure 1 materials-13-03454-f001:**
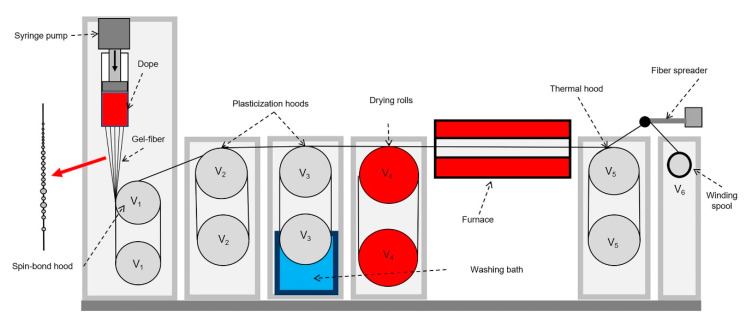
Scheme of the mechanotropic spinning line. V_0_ is the outflow velocity from the die; V_1_–V_6_ are the rotation speeds of the rolls.

**Figure 2 materials-13-03454-f002:**
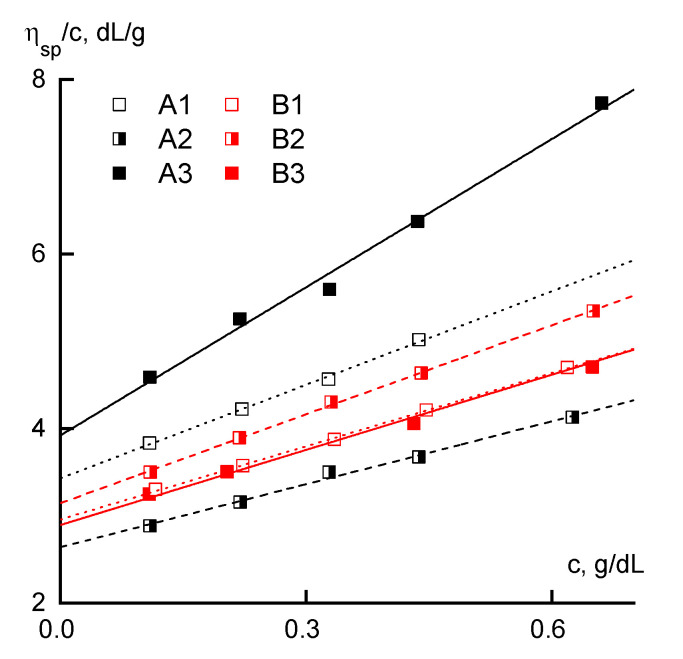
The dependence of the reduced viscosity on the concentration.

**Figure 3 materials-13-03454-f003:**
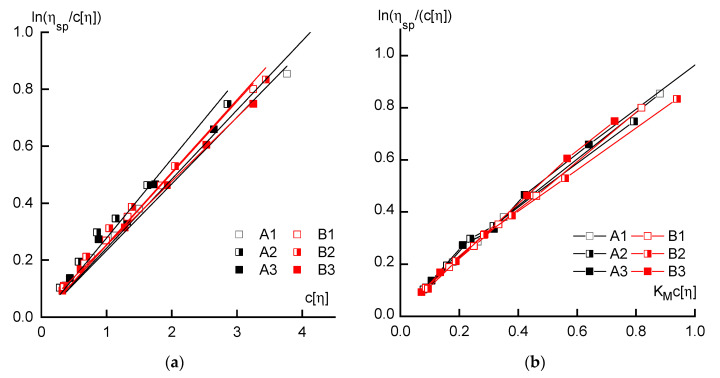
Dependences of reduced viscosity on the volume engaged by macromolecules in the solution (**a**), and the same when using the Martin constant (**b**).

**Figure 4 materials-13-03454-f004:**
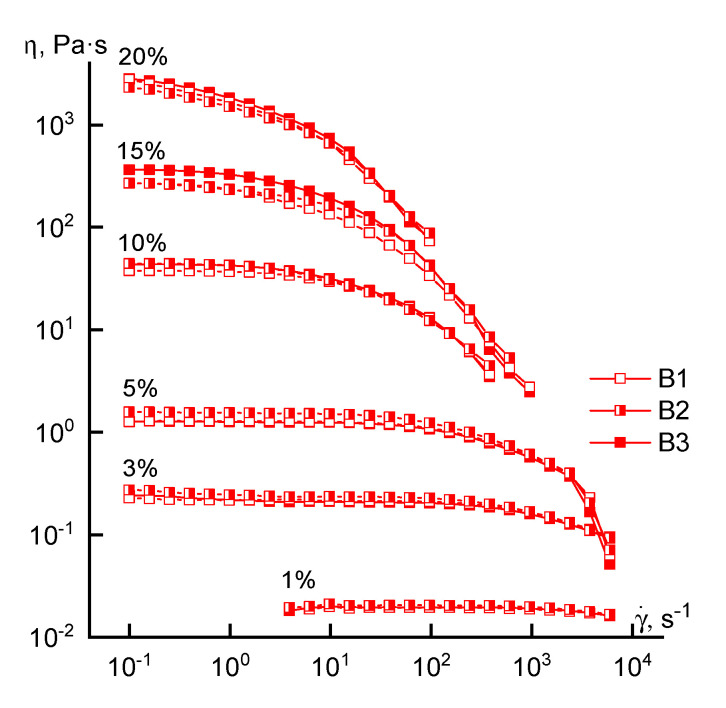
Flow curves of copolymer solutions of various concentrations (series B).

**Figure 5 materials-13-03454-f005:**
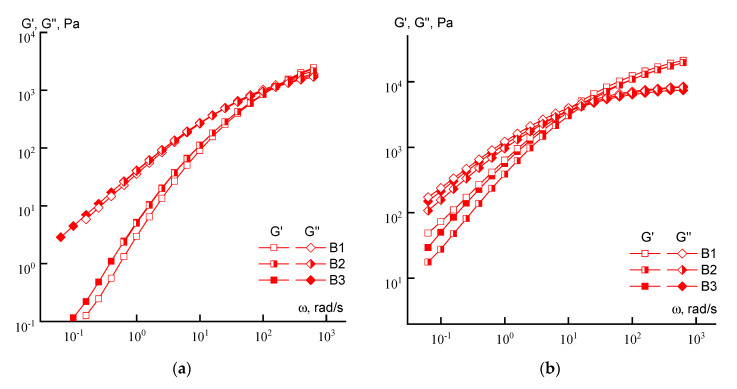
Dependences of the elastic and losses moduli on the frequency of copolymer solutions (series B) with a concentration of 10 (**a**) and 20% (**b**).

**Figure 6 materials-13-03454-f006:**
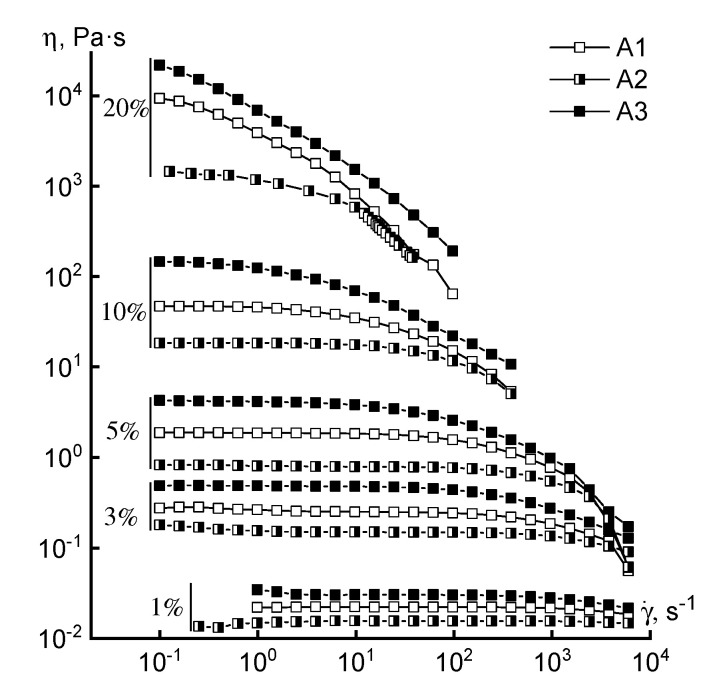
Flow curves of copolymer solutions (series A) of various concentrations (indicated in graphs).

**Figure 7 materials-13-03454-f007:**
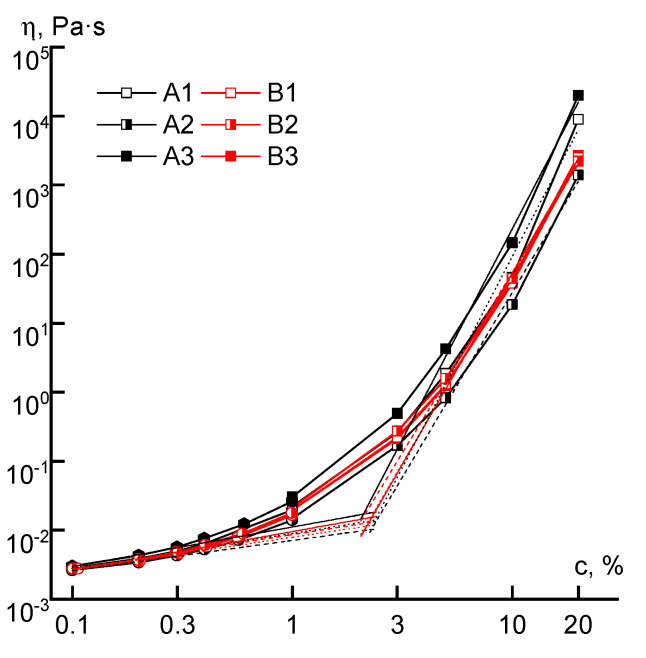
The dependence of the viscosity of series A and B copolymer solutions, measured at a shear rate of ~0.1 s^−1^, on the concentration.

**Figure 8 materials-13-03454-f008:**
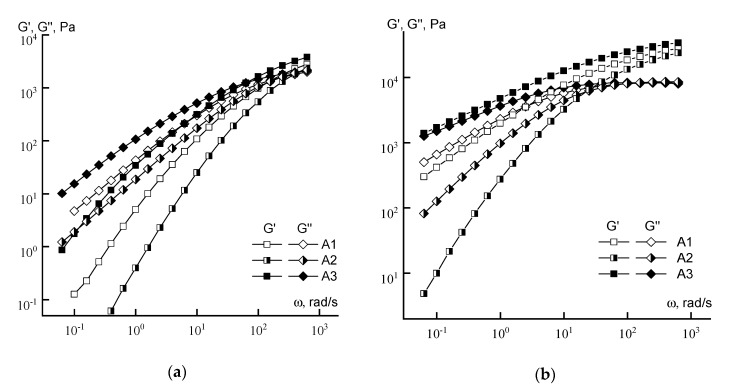
Frequency dependences of the moduli of 10 (**a**) and 20% (**b**) copolymer solutions (series A).

**Figure 9 materials-13-03454-f009:**
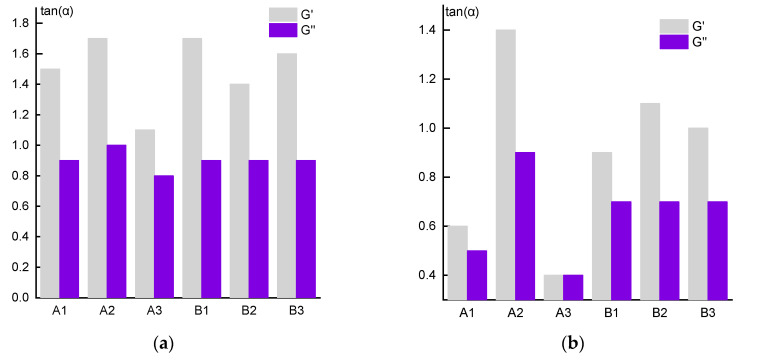
The slope values of the G ’and G’’ frequency dependences for the 10% solutions (**a**) and 20% solutions (**b**) in the region of segmental mobility.

**Figure 10 materials-13-03454-f010:**
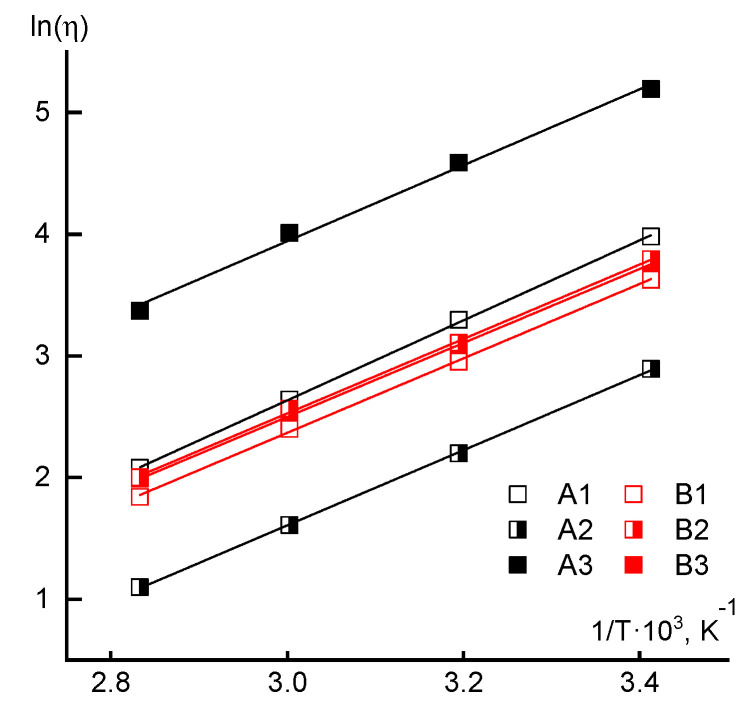
Temperature dependences of the Newtonian viscosity in the coordinates of the Arrhenius equation.

**Figure 11 materials-13-03454-f011:**
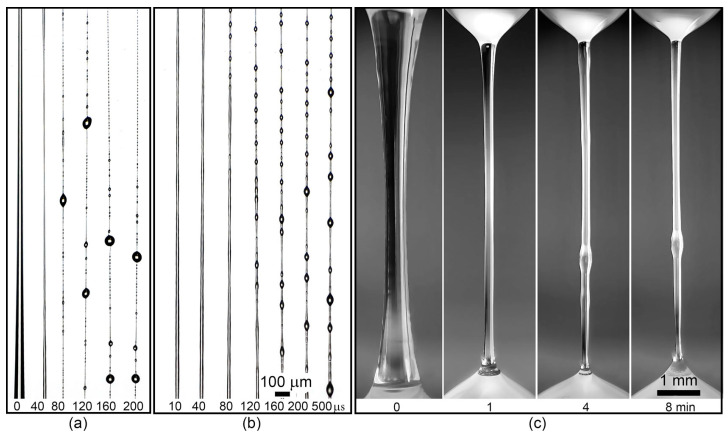
The thinning stages of the A1 copolymer solution drops of different concentrations: (**a**)—10%, (**b**)—15%, (**c**)—20%.

**Figure 12 materials-13-03454-f012:**
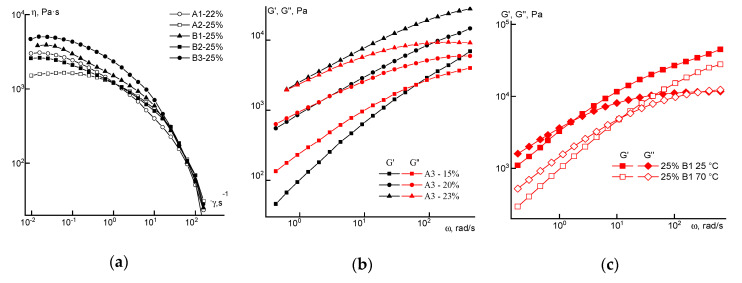
Flow curves (**a**) and frequency dependences (**b**,**c**) of dynamic moduli for solutions of the selected copolymers.

**Figure 13 materials-13-03454-f013:**
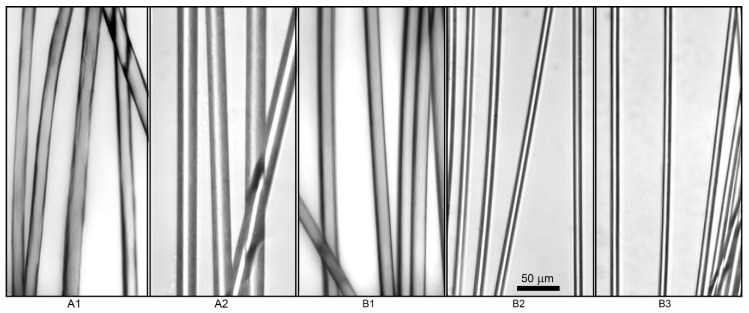
Microphotographs of fibers obtained from PAN solutions. (**A1**) RAFT, continuous; (**A2**) RAFT, semi-batch; (**B1**) FRP, continuous; (**B2**) FRP, semi-batch; (**B3**) FRP, batch. Scale bar is equal for all samples.

**Table 1 materials-13-03454-t001:** Characteristics of the synthesized polymers.

Sample	The Method of AA Addition	M_w_ × 10^−3^	Ð	F_AA_, mol.%
A1	Continuous	176	1.80	10.8
A2	Semi-batch	120	1.54	7.8
A3	Batch	152	1.62	9.8
B1	Continuous	153	2.02	12.4
B2	Semi-batch	175	2.40	13.3
B3	Batch	145	2.14	14.8

**Table 2 materials-13-03454-t002:** Molecular characteristics of the synthesized polymers.

Sample	[η], dL/g	K_M_	K_H_	M_w_ × 10^−3^ g/mole
A1	3.4	0.23	0.30	176
A2	2.6	0.28	0.36	120
A3	4.0	0.24	0.33	152
B1	3.0	0.25	0.32	153
B2	3.1	0.25	0.34	175
B3	2.9	0.23	0.34	145

**Table 3 materials-13-03454-t003:** Crossover frequencies.

Solution Concentration	Crossover Frequency, rad/s					
	A1	A2	A3	B1	B2	B3
10%	158	400	63	180	158	158
20%	2.5	20	0.06	10	17	10

**Table 4 materials-13-03454-t004:** Kinematic parameters for the fiber spinning with polyacrylonitrile and its copolymers (PAN).

Sample	V_0_, m/min	V_1_, m/min	V_2_ m/min	V_3_ m/min	V_4_ m/min at 100 °C	V_5–_V_6_, m/min
A1	0.001	3.8	4	4.5	5	10.2
A2		2.2	2.9	3.7	4.3	6.8
B1		3.8	9.1	9.6	10	20.7
B2		1.6	14.3	16.3	17.3	31.5
B3		2.2	10.6	12.9	14.9	28.5

**Table 5 materials-13-03454-t005:** Mechanical properties of fibers obtained from the copolymer solutions of series A and B.

Sample	Strength, MPa	Elongation at Break, %	Modulus of Elasticity, GPa	Diameter, µm	Draw Ratio
A1	500 ± 42	28 ± 42	4.5 ± 0.3	16 ± 1.5	2.7
A2	560 ± 31	22 ± 1	7.3 ± 0.4	12.6 ± 1	3.1
A3	Spinning unstable				
B1	640 ± 35	27 ± 2.5	4.5 ± 0.2	16 ± 1	5.4
B2	550 ± 23	23 ± 1.9	7.1 ± 0.4	8 ± 1	19.7
B3	710 ± 38	22 ± 1.1	6.7 ± 0.5	8 ± 1	13

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
