# Peer review of "The Effect of the Synthetic Procedure of Acrylonitrile–Acrylic Acid Copolymers on Rheological Properties of Solutions and Features of Fiber Spinning"

_materials, 2020, doi:10.3390/ma13163454_

Round 1
Reviewer 1 Report
See attached document

Author Response
First of all we would like to express our thanks to reviewers who have made pretty important comments which we address below. The revision of the paper has been done according to these comments and all the changes in the revised text are marked with the red color. More detailed information about polymer synthesis and the differences between the samples of series A and B, monomer reactivity, molecular weight distribution and copolymer composition throughout polymerization process is now given in Supplementary Materials (SM) file.
As it was indicated in Materials and Methods, the difference in the copolymers comes from the different methods of their synthesis. We agree that more detailed information about polymer synthesis is required. Aiming this, we provide the manuscript with the file with Supplementary Materials (SM). As it was suggested by referee, this file includes the brief description of RAFT polymerization mechanism, the more detailed analysis of the differences of the samples obtained via different mechanisms and different methods of addition of acrylic acid in polymerization. This analysis is supported by MWDs data (transformation of MWDs throughout polymerization for all the samples of series A and B), dependences of number average MW and dispersity versus monomer conversion, analysis of the change of molar fraction of acrylic acid in copolymer during polymerization, and determination of monomer reactivities. Corresponding reference to the file with SM is added to the Materials and Methods. We would like to note that detailed analysis of the RAFT copolymerization of acrylonitrile and acrylic acid and the influence of the polymerization mechanism and the method of addition of acrylic acid on thermal behavior of the copolymers may be found in ref. 6 and 7.
Concerning macromolecules microstructure, we have shown recently that PAN and its ternary copolymers with methyl acrylate and itaconic acid obtained via conventional radical and RAFT polymerization are atactic and are characterized by similar ratio of triads rr, mm and rm (E.V. Chernikova, R.V. Toms, N.I. Prokopov, V.R. Duflot, A.V. Plutalova, S.A. Legkov, and V.I. Gomzyak; Thermooxidative Stabilization of Acrylonitrile Terpolymers Obtained under Reversible Chain-Transfer Conditions: Effects of Synthesis Temperature and Initiation Method. Polym. Sci. Ser. B, 2017, 59, 28-42, doi: 10.1134/S156009041701002X). We suppose that atactic microstructure is kept in binary copolymers of acrylonitrile and acrylic acid.
Reviewer 2 Report
The paper entitled ‘Effect of synthetic procedure of acrylonitrile-acrylic acid copolymers on rheological properties of solutions and features of fiber spinning’ is an interesting study focusing on rheological properties of solutions, the stability of electrospinning and the properties of the resulted fibers. This manuscript is related to polymers therefore I wonder why it was not submitted to the Polymers (MDPI) for example. Anyway, as polymers are also materials and authors show the complex theoretical and experimental comparison between different spinning techniques from theoretical perspectives it is suitable for Materials (MDPI) too and deserve to be published. Here are a few minor comments:
- Country of polymers is missing in the materials section
- What was the load cell used in mechanical testing, and what was RH during the tests?
- Why the bullet point are used in the lines 132-138? Also not sure if the bullet points are necessary in lines 158-164.
- The description of the statistical analysis is missing? How was the error calculated?
- Could you provide more specific names/symbols for samples instead of A1, B1?
- Could you provide units in [..] in all graphs and tables?
- Line 361, 371 … instead of ‘microns’, you could use µm.
- Line 437 removed the?
Author Response
First of all we would like to express our thanks to reviewers who have made pretty important comments which we address below. The revision of the paper has been done according to these comments and all the changes in the revised text are marked with the red color. More detailed information about polymer synthesis and the differences between the samples of series A and B, monomer reactivity, molecular weight distribution and copolymer composition throughout polymerization process is now given in Supplementary Materials (SM) file.
Comment 1. Country of polymers is missing in the materials section.
Response. The polymers have been synthesized ourselves. However, we have carefully checked materials section and extended the information about suppliers of the reagents used.
Comment 2. What was the load cell used in mechanical testing, and what was RH during the tests?
Response. The fiber sample of 25 mm length kept in standard pneumatic clamps of extension machine Instron 1122 with the working part of 10 mm. The relative humidity of environment atmosphere does not exceed 40%.
Comment 3. Why the bullet point are used in the lines 132-138? Also not sure if the bullet points are necessary in lines 158-164.
Response. It was fixed.
Comment 4. The description of the statistical analysis is missing? How was the error calculated?
Response.
The experimental data exceeded 3σ were excluded from analysis. Confidence interval was estimated with a 95% confidence level. So standard deviation was calculated as . The sample average is calculated as: , where n is a sample of observations and σ is a sample standard deviation. It was defined as the square root of the sample variance: . This information was added into Experimental.
Comment 5. Could you provide more specific names/symbols for samples instead of A1, B1?
Response. No, we have given a full description of the samples in the Experimental part
Comment 6. Could you provide units in [..] in all graphs and tables?
Response. No, in the rules of the journal there are no special requirements for the designation of dimensions
Comment 7. Line 361, 371 … instead of ‘microns’, you could use µm.
Response. Fixed
Comment 8. Line 437 removed the?
Response. Fixed
Reviewer 3 Report
The manuscript by Skvortsov et al. presents an interesting study on the effect of the synthetic procedure in the rheological behavior and spinning process of copolymer of acrylonitrile and acrylic acid. The work is well done and the results are sound. I have only a minor concerning:
-It would be nice if the author coul present some information about the monomer distribution obtained using some ancillary technique.
Author Response
First of all we would like to express our thanks to reviewers who have made pretty important comments which we address below. The revision of the paper has been done according to these comments and all the changes in the revised text are marked with the red color. More detailed information about polymer synthesis and the differences between the samples of series A and B, monomer reactivity, molecular weight distribution and copolymer composition throughout polymerization process is now given in Supplementary Materials (SM) file.
Response. We agree with this comment and provide the manuscript with the file with Supplementary Materials. In particular, this file contains information about monomer reactivities and dependences of the molar fraction of acrylic acid from overall monomer conversion for all the samples as well as detailed description of the differences of the samples obtained by different mechanisms and different methods of addition of acrylic acid into reaction. The corresponding reference to this file is given in the Materials section.